

# The influence of parenting style and coping behavior on nonsuicidal self-injury behavior in different genders based on path analysis

Fang Cheng[1], Changzhou Hu[1], Wenwu Zhang[1], Huabing Xie[2], Liangliang Shen[3], Beini Wang[1], Zhenyu Hu[1], Yucheng Wang[1] and Haihang Yu[1]

[1] Department of Pediatric Psychology, Ningbo Kangning Hospital, Ningbo, China
[2] Department of General Medicine, People's Hospital of Wuhan University, Wuhan, China
[3] School of Chemical Engineering, Dalian University of Technology, Dalian, China

## ABSTRACT

**Background:** Nonsuicidal self-injury (NSSI) behaviors—an important factor that profoundly affects the physical and mental health of young people—are induced by complex and diverse factors, while showing significant differences at the gender level. We examined mediating behaviors among parenting styles, students' coping styles, and endogenous and exogenous influencing variables of adolescents' NSSI behaviors.
**Methods:** In this cross-sectional study, Secondary school students in Ningbo, Zhejiang Province, China ($n = 2,689$; F/M:1532/1157) were surveyed for basic attributes, parenting styles, coping styles, and NSSI behaviors. After the initial screening of the sample data, several external derivatives were screened based on the single factor analysis method. On this basis, the construction of path analysis models under multivariate multiple elicitations was carried out.
**Results:** The overall prevalence of NSSI was 15.16%, and the incidence of NSSI in boys was lower than that in girls (OR = 0.334, 95% CI [0.235–0.474]). The path analysis model data fit well; the indicators of female and male part are: CFI = 0.913/0.923, GFI = 0.964/0.977, SRMR = 0.055/0.047, RMSEA = 0.097/0.069 with 90% confidence interval (CI) [0.084–0.111]/[0.054–0.084]. For female, when negative coping style and extreme education affect NSSI respectively, the standardized path coefficient values are 0.478 (z = 20.636, P = 0.000 < 0.01) and 0.151 (z = 6.524, P = 0.000 < 0.01) respectively, while for male, the corresponding values become 0.225 (z = 7.057, P < 0.001) and 0.104 (z = 3.262, P < 0.001).
**Conclusion:** In particular, we investigated the mediating effects of gender-specific NSSI influences and found that NSSI behaviors were strongly associated with environmental variables and individual factors, especially family parenting style and adolescent coping style, which influenced NSSI in a gender-specific manner. The results showed that males were the target of both positive and negative parenting styles, whereas females were more likely to choose negative coping styles directed towards emotions in response to external stimuli, and instead showed a more significant predisposition towards NSSI behaviors. This phenomenon seems to be influenced by multilevel factors such as sociocultural, individual value identity, and physiological structure differences. In the path analysis model with the introduction

Corresponding author
Haihang Yu,
yuhaihang0414@sina.com

of mediating effects, the influence of gender differences on NSSI behavior becomes more pronounced under the interaction of multiple factors: women seem to be more significantly influenced by the external derivatives in the internal derivatives than male subjects, and are more likely to trigger NSSI behavior under the interaction of multiple factors. These findings effectively reveal the significant role of different end-influencing factors in NSSI behaviors at the level of gender differences, which can provide effective theoretical support to prevent and treat NSSI behaviors in adolescents.

## INTRODUCTION

Non-suicidal self-injury (NSSI) is the intentional, direct injury to body tissues without suicidal intent—an act that is socially and culturally unacceptable. Cutting is the most predominant form of NSSI, and other injury types include burns, scrapes or scratches to the skin, and bites (*ISSS, 2022*). The incidence of NSSI is highest in the adolescent population, usually between the ages of 12–15 years, with a peak in mid-adolescence (*Plener et al., 2015*; *Cipriano, Cella & Cotrufo, 2017*). The complex social causes and etiology of NSSI make the prevalence of this behavior vary significantly across countries. Data show that the global prevalence of NSSI in adolescents is about 17.2% (*Swannell et al., 2014*). The prevalences of NSSI in adolescents in Scotland, the United States, and Germany were 13.8%, 15.3%, and 3.1%, respectively, while in China the prevalence was about 22.4% (*Lang & Yao, 2018*). Although NSSI is a maladaptive but usually effective coping strategy for managing emotions in adolescents, and its attenuated distress may be followed by attenuated NSSI behavior, the behavior may be persistent in a large number of young people, especially females (*Hawton, Saunders & O'Connor, 2012*), with probabilities ranging from 17.1% to 38.6% (*Brunner et al., 2014*). Furthermore, extensive evidence has shown that NSSI in adolescence is strongly associated with concurrent and subsequent suicidal ideation or attempts (*Plener et al., 2015*; *Koenig et al., 2017*; *Mars et al., 2019*), psychological symptoms (*e.g.*, depression, anxiety, borderline personality disorder) (*Baetens et al., 2014*; *Wang et al., 2022*), and other psychosocial dysfunction (*e.g.*, cognitive vulnerability, sequelae of childhood sexual abuse, interpersonal distress) (*Kaess et al., 2021*; *Gu, Ma & Xia, 2020*). At the same time, as a behavior strongly related to social factors, it is significant for us to deeply explore the multi-factor interactive relationship between NSSI behavior at the social family level and individual psychological behavior level and to make effective preventive measures. Therefore, NSSI has gradually evolved into a prominent public health problem for adolescents worldwide, and an investigation into the causes and influences of NSSI deserves attention.

The etiology of NSSI is complex and dynamic, involving genetic, biological, spiritual, psychological, physical, social, and cultural crossover effects (*Kaess et al., 2021*; *Richmond-Rakerd et al., 2019*). Its motivation can be divided into intrapersonal functions (including emotion regulation, thought regulation, and self-punishment) and

interpersonal functions (including conveying distress, social influence, and punishment) (*Taylor et al., 2018*). Adolescence is a stage of gradual psychological maturity, and adolescents have not formed complete values and judgment at this time, thus a variety of psychological problems can occur. The inability to regulate their emotions and cope well with stress can lead to emotional disorders such as depression and anxiety, and then self-injurious behavior. The family environment plays an important role in shaping the personality and behavioral characteristics of adolescents. There is evidence that parenting styles, coping behaviors, and gender have significant effects on NSSI behaviors (*Wang et al., 2022*; *Fan et al., 2021*). The parent-child relationship is the primary relationship a child has, and the health of the parent-child relationship influences the child's social and emotional development (*Wang et al., 2022*; *Gruhn & Compas, 2020*). Parents are the primary agents of child socialization and major contributors to the health and development of adolescents. Some parenting characteristics such as parenting style are highly correlated with the occurrence of NSSI in adolescents (*Gu, Ma & Xia, 2020*; *Serafini et al., 2017*). Negative parenting styles are a significant risk factor for NSSI (*Guerreiro et al., 2013*; *Victor et al., 2019*; *Van Lissa et al., 2019*; *Zanarini, 2009*; *Chen et al., 2022*). Attachment theory (*Victor et al., 2019*; *Zanarini, 2009*; *Bussey & Bandura, 1999*) suggests that parenting behaviors play different roles in the cognitive and emotional development of children during adolescence. *Crowell, Beauchaine & Linehan*'s *(2009)* biosocial model suggests that if a biased form of family parenting is experienced during childhood, emotional experiences are denied or neglected, which can inhibit the acquisition of emotion regulation skills and thus predispose individuals to negative coping styles, for instance, NSSI.

A positive coping style oriented to the way of coping with stressful events, and circumstances, and the ability to regulate emotions play an important role in the development of resilience and the reduction of psychopathological symptoms during childhood and adolescence (*Panlilio, Jones Harden & Harring, 2018*). The dimensions of emotional dysregulation most closely related to NSSI include difficulties with impulse control, limited regulatory strategies, unacceptable emotional responses, and difficulties with goal-directed behavior (*Wolff et al., 2019*). *Guerreiro et al. (2013)* found that NSSI in adolescents was associated with negative coping styles, and this was confirmed in a study by *Chu, Pan & Dong (2012)*. Adolescents with NSSI were more likely to adopt avoidant coping styles compared to those without NSSI (*Williams & Hasking, 2010*; *Cawood & Huprich, 2011*; *Kiekens et al., 2015*; *Castro & Kirchner, 2018*). *Castro & Kirchner (2018)* observed 965 adolescents and found that adolescents who chose emotionally directed coping styles such as avoidance were 3.5 times more likely to engage in NSSI behaviors than those who adopted problem-solving styles. The above studies suggest that negative coping styles such as avoidance, self-blame, venting, patience, behavioral disengagement, and fantasy increase the risk of NSSI, but the effect of positive coping styles on NSSI in adolescents is unclear (*Castro & Kirchner, 2018*; *Lin et al., 2017*; *Wan et al., 2020*). At the same time, there is an influential relationship between parenting styles and coping styles; positive parenting styles such as warmth and understanding, promote mature, positive coping styles in adolescents while negative parenting styles such as punishment and

rejection promote immature, negative coping styles (*Wan et al., 2015*; *Fang & Li, 2019*; *Chen, Zhao & Chen, 2017*). Scholars have discussed in depth the various influencing factors of NSSI, but there seems to be a lack of study on the interaction of these factors on NSSI behavior. Therefore, it would be interesting to investigate various factors such as personal and external variables to understand the relationship between parenting styles and NSSI behaviors.

In particular, men and women differ in gender-specific behaviors of NSSI because of different influencing mechanisms including physiological, psychological, and social factors. Current research on gender differences in NSSI behavior focused on morbidity and self-injurious behavior. The effect of gender on the incidence of NSSI has been studied with varying results. *Swannell et al. (2014)* showed that NSSI is not affected by gender. However, studies by *Plener et al.(2015)* and *Monto, McRee & Deryck (2018)* did show a small increase in the risk of NSSI in females (*Plener et al., 2015*; *Monto, McRee & Deryck, 2018*; *Wilkinson et al., 2022*). Studies by *Wilkinson et al. (2022)* showed that the effect of gender on NSSI varies by age; in females, the incidence of NSSI increases from early adolescence, peaks in mid-adolescence, and then gradually decreases; in males, the NSSI remains at similar levels at all ages (*Plener et al., 2015*; *Wilkinson et al., 2022*). In terms of the behavioral effects of gender on NSSI, adolescent girls are more likely to use methods involving blood (especially cutting and scratching), whereas adolescent boys are more likely to beat themselves, burn themselves, or bang their heads (*Bresin & Schoenleber, 2015*; *Sornberger et al., 2012*). These studies show that girls are more likely to have an NSSI, but they do not address why females are more likely than males to engage in an NSSI. As with all complex behaviors, there may be multiple factors at play. The study of the interaction of each influencing factor in the gender difference of NSSI deserves attention.

Although scholars have discussed in depth the influencing factors of NSSI in various aspects, few reports have been able to effectively investigate the mechanisms of NSSI-influencing factors in-depth due to the different mechanisms of influencing effects and the different direct and indirect contributions of feedback to NSSI behavior. Therefore, in order to delve deeper into the impact mechanism of NSSI behavior and its prevention guidance, in this work, we propose a path analysis model-based NSSI relational network model to effectively capture the distal and proximal effects of various dimensions of gender differences on NSSI behaviors under a multifactorial interactive influence mechanism. A cross-sectional survey of 2,769 cases was conducted to investigate the influence mechanism and contribution of NSSI based on numerical analysis and conceptual modeling, which effectively reveals the intrapersonal and interpersonal factors underlying gender differences in NSSI, and can provide strong theoretical guidance for effective prevention and response measures of NSSI.

# METHOD

## Subjects

A cross-sectional survey was carried out in Ningbo City, China from September to October 2021. The study population was derived from three junior middle schools and three senior high schools in four districts of Ningbo City (Haishu, Jiangbei, Gaoxin, and Yinzhou).

Specific investigation process: students from grade 1 to grade 3 of junior middle school and grade 1 to grade 2 of senior high school were randomly selected. After a trained psychiatrist reads out the instructions, all students complete the questionnaire on-site within 40 min.

We obtained the written informed consent of all participants and their parents before the study, including the potential trigger nature of all relevant issues specifically mentioned, as well as confidentiality restrictions related to mandatory reporting concerns. The study was approved by the ethics committee of Ningbo Kangning hospital (Ethical Application Ref. No.: nbknyy-2020-Lc-52), which was approved in October 2020 and is valid until October 2023. All research investigation methods carried out in this work were conducted in accordance with the relevant guidelines and regulations in the declaration of Helsinki.

## Survey content

Data on demographic characteristics and conditions considered to be related to NSSI were collected (*Lin et al., 2017*; *Wan et al., 2015*). This included age, gender, physical condition, academic achievement, peer relationship, family relationship, family economic situation, parental education level, and other factors. In particular, the revised coping style scale for middle school students and the parental rearing style evaluation scale (EMBU) jointly prepared by C. Perris, Department of Psychiatry, Umea University, Sweden, were used to effectively evaluate the individual and interpersonal factors of the subjects.

### Basic characteristics

We adapted questions on the basic characteristics of participants from a study by *Fang & Li (2019)* on NSSI behavior and risk factors of patients with depression. The following parameters in the basic status questionnaire were included: age (11–13, 14–16), gender (male and female), physical status (weak, average, good), academic performance (poor, average, good, excellent), peer relationship (poor, average, close), parental relationship (separation/divorce, domestic violence, quarrel conflict, harmony), *per capita* monthly income of the family (<3,000, 3,000–6,000, 6,000–10,000, >10,000), *per capita* housing area of the family (<20, 20–40, 40–60, >60), parents' investment in their children (very low, low, average, high), parents' expectations for their children (none, independent life, range oneself, attend university), fathers' and mothers' education level (primary school, junior middle school, senior high school, university and above).

### Students' coping styles

Coping styles refers to the cognitive and behavioral efforts made by individuals to meet their endogenous and external needs, including positive coping and negative coping (*Chen, Zhao & Chen, 2017*), which usually include (1) problem-solving, seeking social support, seeking help and rationalization, and (2) emotional coping styles such as patience, retreat, venting emotions and fantasies, denial, and so on.

We used a middle school students' Coping Styles Scale to measure the subjects' coping styles in the face of events (Cronbach α: 0.84). There are 36 items on the survey scale, and each item is graded according to the four-level scoring standard of adoption, occasional

adoption, occasional adoption, and frequent adoption. The subscale consists of two parts: (1) problem-oriented coping subscale, which includes three factors: P1 problem-solving, P2 seeking social support, and P3 positive rationalization explanation; (2) emotional-oriented coping subscale, which includes four factors: E1 avoidance, E2 escape, E3 venting, and E4 fantasy denial.

### Parenting styles

The parenting styles evaluation scale (EMBU) was introduced to effectively evaluate the relationship between parenting style and individual NSSI behavior. The scale contains 66 items, of which (1) the subscale of paternal rearing style contains six factors, which are F1 emotional warmth and understanding, F2 severe punishment, F3 refusal and denial, F4 preference for subjects, F5 excessive interference, and F6 excessive protection; (2) The subscale of maternal rearing style includes five factors: M1 emotional warmth and understanding, M2 severe punishment, M3 refusal and denial, M4 preference for subjects, and M5 excessive interference and protection. The Cronbach $\alpha$ values of the parental EMBU subscale are 0.86 and 0.89 respectively.

### Non-suicidal self-injury behavior

NSSI behavior is defined according to DSM-5 diagnostic criteria, and the following true-or-false questions are designed: (1) In the past year, there have been more than five acts of intentional injury to oneself that are not to end life and cause mild or moderate physical injuries such as bleeding, contusion, and pain; (2) The above behaviors have the purpose of extricating from a tense feeling or cognitive state and/or solving interpersonal difficulties and/or inducing a positive emotional state. Answering "yes" for the above two questions indicates that there is NSSI behavior.

## Quality control

To ensure the reliability and consistency of the questionnaire results, psychiatrists administered the test to participants. A pre-test was conducted before the survey. The purpose of the survey was explained on-site. After the test, the doctors of the main test will collect the questionnaire, enter and screen the questionnaires from two people, and then conduct a logical check on the questionnaire to ensure the quality of the answers. A total of 2,784 questionnaires were administered and 2,753 questionnaires were recovered, with a recovery rate of 98.89%. Among them, 64 questionnaires had >30% of missing data and logical errors; thus were recorded as invalid questionnaires. There were 2,689 valid questionnaires, and the effective rate was 96.59%.

## Data analysis method

After logic checking and proofreading, Epidata 3.1 Software was used for data entry. The scores of NSSI behavior, parental rearing style, and coping style of male and female students were compared by independent sample t-test; The count data were described by the number of cases (constituent ratio) method. Comparisons between groups were made using the $\chi^2$ test. All analyses were conducted with SPSS software, version 26.0.

## RESULTS

### Single-factor analysis

A total of 2,689 participants were investigated, among which 435 participants had NSSI behavior, and the positive rate was 15.16%. The numbers of participants by age group were: 11–13 years old: 817 (97 in the NSSI group, 11.87%); 14–16 years old: 1,872 (338 in the NSSI group, 18.06%). Gender characteristics of the sample: male: 1,157 (78 in NSSI group, 6.74%); female: 1,532 (357 in the NSSI group, 23.30%). It was found that the age and sex of middle school students were statistically significant ($P < 0.05$).

#### Single-factor research based on basic characteristics

To further study the single factor influencing the relationship between the basic characteristics of the survey sample and NSSI behavior, we investigated the relationship between the occurrence of NSSI and physical condition, academic achievement, peer relationship, parent relationship, family *per capita* monthly income, family *per capita* housing area, parents' expectations for children, parents' investment in children (money and time), and parent's education level. The statistical results are shown in Table 1. We found no significant difference in the occurrence of NSSI between middle school students and parents' expectations and investment in children ($P > 0.05$); however, in the comparison of differences between NSSI behavior and other basic characteristics of middle school students, the results of the univariate analysis with statistical significance are proposed in this part ($P < 0.05$).

#### Gender differences between parenting styles and coping styles

The research shows that NSSI is a social behavior under the synergistic action of multiple factors, which is not only affected by the basic characteristics of the participant but also closely related to the participants' family rearing style and coping style. Therefore, a study based on parental rearing styles and coping styles of middle school students will be deeply discussed in this work.

Table 2 shows the significant factor scores of parenting style, NSSI behavior, and gender respectively. Compared with middle school students with NSSI behavior, the non-NSSI group experienced a more significant positive parenting style: the score of the emotional warmth understanding factor was significantly higher ($P < 0.05$). However, middle school students with NSSI behavior experienced relatively frequent negative parenting styles: severe punishment, refusal and denial, excessive interference, and excessive protection, with high scores, and the differences are statistically significant ($P < 0.05$). At the same time, a gender-based difference in parental rearing styles was found: whether it is positive rearing styles such as emotional warmth and understanding, or negative rearing behaviors such as severe punishment, refusal and denial, excessive intervention, and excessive protection, the scores of rearing behavior factors for men are higher than those for women, and the difference is statistically significant ($P < 0.05$). Overall, the scores of caring education factors in the non-NSSI group were higher, and the scores of radical education in the NSSI group were higher ($P < 0.05$). Compared with the female group, the score

**Table 1 Single-factor analysis of influencing factors of NSSI in middle school students.**

| Variables (tertile score) | | No NSSI (%) | NSSI (%) | Z-Value | p-Value |
|---|---|---|---|---|---|
| Physical condition | Weak | 46 (73.02) | 17 (26.98) | 8.12 | <0.001 |
| | Normal | 405 (73.24) | 148 (26.76) | | |
| | Good | 1,803 (86.98) | 270 (13.02) | | |
| Academic performance | Poor | 204 (71.33) | 82 (28.67) | 4.27 | <0.001 |
| | Average | 831 (83.27) | 167 (16.73) | | |
| | Good | 1,018 (88.44) | 133 (11.56) | | |
| | Excellent | 201 (79.13) | 53 (20.87) | | |
| Peer relationship | Poor | 23 (37.10) | 39 (62.90) | 7.27 | <0.001 |
| | Average | 1,164 (81.97) | 256 (18.03) | | |
| | Close | 1,067 (88.40) | 140 (11.60) | | |
| Parental relationship | Separation/divorce | 134 (66.33) | 68 (33.67) | 14.01 | <0.001 |
| | Domestic violence | 19 (46.34) | 22 (53.66) | | |
| | Quarrel conflict | 261 (67.62) | 125 (32.38) | | |
| | Harmony | 1,840 (89.32) | 220 (10.68) | | |
| Monthly per capita income (Yuan) | <3,000 | 103 (88.03) | 14 (11.97) | 7.61 | <0.001 |
| | 3,001~6,000 | 717 (90.19) | 78 (9.81) | | |
| | 6,001~10,000 | 858 (84.37) | 159 (15.63) | | |
| | >10,000 | 576 (75.79) | 184 (24.21) | | |
| Per capita household housing area (m$^2$) | <20 | 167 (87.89) | 23 (12.11) | 6.81 | <0.001 |
| | 21~40 | 884 (88.67) | 113 (11.33) | | |
| | 41~60 | 545 (84.50) | 100 (15.50) | | |
| | >60 | 658 (76.78) | 199 (23.22) | | |
| Parents' investment in their children (time and money) | Very low | 38 (74.51) | 13 (25.49) | 0.42 | 0.67 |
| | Low | 183 (79.57) | 47 (20.43) | | |
| | Average | 1,414 (85.70) | 236 (14.30) | | |
| | High | 619 (81.66) | 139 (18.34) | | |
| Parents' expectations of their children | None | 22 (43.14) | 29 (56.86) | 1.69 | 0.90 |
| | Independent life | 425 (87.27) | 62 (12.73) | | |
| | Marry and settle down | 293 (81.39) | 67 (18.61) | | |
| | Attend university | 1,514 (84.53) | 277 (15.47) | | |
| Educational level of the father | Primary school | 329 (52.14) | 302 (47.86) | 19.09 | <0.001 |
| | Junior high school | 1,212 (93.81) | 80 (6.19) | | |
| | High school | 455 (93.81) | 30 (6.19) | | |
| | University and above | 258 (91.81) | 23 (8.19) | | |
| Educational level of the mother | Primary school | 459 (59.84) | 308 (40.16) | 17.45 | <0.001 |
| | Junior high school | 1,090 (93.56) | 75 (6.44) | | |
| | High school | 436 (93.97) | 28 (6.03) | | |
| | University and above | 269 (91.81) | 24 (8.19) | | |

**Table 2 Significant factors of parenting style, NSSI behavior, and gender.**

| Parenting style | | No NSSI | NSSI | t-Value | p-Value | Male | Female | t-Value | p-Value |
|---|---|---|---|---|---|---|---|---|---|
| Father's parenting style | F1 | 52.08 ± 11.99 | 40.16 ± 11.07 | 20.29 | <0.001 | 51.34 ± 12.24 | 49.26 ± 12.85 | 4.25 | <0.001 |
| | F2 | 19.61 ± 7.33 | 22.80 ± 9.09 | 6.89 | <0.001 | 21.24 ± 7.81 | 19.29 ± 7.57 | 6.5 | <0.001 |
| | F3 | 10.34 ± 3.78 | 11.90 ± 4.25 | 7.1 | <0.001 | 11.18 ± 3.94 | 10.15 ± 3.82 | 6.78 | <0.001 |
| | F4 | 20.73 ± 5.05 | 22.70 ± 6.24 | 6.22 | <0.001 | 21.91 ± 4.99 | 20.39 ± 5.45 | 7.38 | <0.001 |
| | F5 | 12.63 ± 3.52 | 13.20 ± 3.81 | 2.92 | 0.002 | 13.25 ± 3.61 | 12.32 ± 3.49 | 6.71 | <0.001 |
| | F6 | 72.86 ± 17.63 | 82.88 ± 19.99 | −9.753 | 0.002 | 77.40 ± 18.40 | 72.27 ± 18.10 | 7.224 | <0.001 |
| Mother's parenting style | M1 | 53.35 ± 11.91 | 42.66 ± 11.32 | 17.89 | <0.001 | 52.38 ± 12.29 | 51.05 ± 12.54 | 2.77 | 0.006 |
| | M2 | 14.09 ± 5.61 | 16.98 ± 6.43 | 8.77 | <0.001 | 15.09 ± 6.09 | 14.15 ± 5.63 | 4.11 | <0.001 |
| | M3 | 14.24 ± 5.38 | 17.28 ± 5.71 | 10.25 | <0.001 | 15.09 ± 5.64 | 14.46 ± 5.46 | 2.88 | 0.004 |
| | M4 | 36.00 ± 7.97 | 40.58 ± 8.72 | 10.15 | <0.001 | 37.72 ± 7.88 | 36.01 ± 8.48 | 5.35 | <0.001 |
| | M5 | 73.97 ± 17.54 | 87.29 ± 18.48 | −14.37 | <0.001 | 77.84 ± 18.37 | 74.82 ± 18.25 | 4.232 | <0.001 |
| Global factors | | | | | | | | | |
| Positive education | | 105.45 ± 22.97 | 82.84 ± 20.56 | 20.59 | <0.001 | 103.75 ± 23.67 | 100.32 ± 24.29 | 3.67 | <0.001 |
| Radical education | | 146.82 ± 33.53 | 170.17 ± 34.78 | −13.21 | <0.001 | 155.24 ± 35.12 | 147.10 ± 34.17 | 6.048 | <0.001 |

**Note:**
Significant factors of parenting style, NSSI behavior, and gender ($\acute{x} \pm s$).

factors of caring education and radical education in male participants were higher ($P < 0.05$).

Table 3 shows the comparison of significant factor scores of middle school students' coping styles with NSSI behavior and gender. Compared with middle school students with NSSI behavior, the participants in the non-NSSI group were more inclined to adopt problem-solving–oriented positive coping styles: solving problems, seeking social support, and positive rational explanation. At the same time, the emotional coping styles (patience, avoidance, venting, fantasy denial) were in a disadvantageous position, and the differences were statistically significant ($P < 0.05$). In particular, an interesting phenomenon was also found in Table 3, where gender showed obvious differences in the choice of behavior patterns in the face of point problems or emotions. Boys were more inclined to problem-oriented coping styles such as problem-solving and positive rational explanation, while girls were more significantly inclined to adopt emotion-oriented coping behaviors such as escape, venting, and fantasy denial ($P < 0.05$). What's more, there was no significant gender difference in the coping styles of seeking social support and enduring emotions ($P > 0.05$). Overall, the positive coping style factor score of the non-NSSI group was higher, and the negative coping style score of the NSSI group was higher ($P < 0.05$); Compared with the girls, the score factor of the positive coping style of the boys showed a higher level ($P < 0.05$).

## Construction of path model and the mediating effects of NSSI behavior

In our previous article, based on the basic characteristics of the sample, parental rearing style, and middle school students' coping behavior, we carried out a single factor analysis with the NSSI behavior/gender of the sample respectively. The screening excluded the variables with weak statistical significance between parents' investment in children and

**Table 3 Significant factors of coping styles, NSSI behavior, and gender of middle school students.**

| Coping styles | | No NSSI | NSSI | t-Value | p-Value | Male | Female | t-Value | p-Value |
|---|---|---|---|---|---|---|---|---|---|
| Problem-solving–oriented coping | P1 | 21.32 ± 3.83 | 16.33 ± 4.84 | −20.29 | <0.001 | 21.13 ± 4.16 | 20.04 ± 4.53 | 6.47 | <0.001 |
| | P2 | 19.46 ± 4.14 | 15.64 ± 4.38 | −16.78 | <0.001 | 18.79 ± 4.39 | 18.88 ± 4.21 | −0.49 | 0.63 |
| | P3 | 14.49 ± 2.85 | 11.08 ± 3.55 | −18.91 | <0.001 | 14.26 ± 2.99 | 13.70 ± 3.38 | 4.52 | <0.001 |
| Emotional-oriented coping | E1 | 9.65 ± 2.37 | 11.09 ± 2.32 | 11.62 | <0.001 | 9.85 ± 2.45 | 9.91 ± 2.40 | −0.61 | 0.54 |
| | E2 | 7.66 ± 2.23 | 9.53 ± 2.68 | 13.66 | <0.001 | 7.69 ± 2.27 | 8.17 ± 2.49 | −5.17 | <0.001 |
| | E3 | 8.36 ± 2.57 | 9.65 ± 2.83 | 8.87 | <0.001 | 8.11 ± 2.52 | 8.91 ± 2.70 | −7.85 | <0.001 |
| | E4 | 10.27 ± 3.37 | 12.87 ± 3.66 | 13.73 | <0.001 | 10.33 ± 3.38 | 10.96 ± 3.65 | −4.59 | <0.001 |
| Global factors | | | | | | | | | |
| Positive coping style (problem-solving–oriented) | | 55.29 ± 8.95 | 43.06 ± 10.93 | 21.94 | <0.001 | 54.20 ± 9.79 | 52.63 ± 10.68 | 3.963 | <0.001 |
| Negative coping style (emotional-oriented) | | 35.94 ± 7.69 | 43.14 ± 8.07 | −17.174 | <0.001 | 35.99 ± 7.81 | 37.95 ± 8.38 | −6.244 | <0.001 |

**Note:**
Significant factors of coping styles, NSSI behavior, and gender of middle school students ($\bar{x} \pm s$).

parents' expectations for children and NSSI, as well as the variables with weak statistical significance between seeking social support and patience and gender differences. However, the single factor analysis of parameters is a relatively simple causal relationship model. The relationship between independent variables and dependent variables is one-to-one corresponding, and there is no form of interaction between variables. NSSI behavior is a complex individual behavior under the cross-influence of multiple factors, such as social relations, individual psychology, and behavioral coping style. The single factor analysis method is difficult to be used in deeply exploring the interaction and intermediary effects between the influencing factors of NSSI.

Therefore, in this work, we propose a path distribution network of NSSI influencing factors based on a path analysis model. At the same time, the construction of the intermediary effect of NSSI endogenous derivative factors at the level of genders will be discussed in depth. To effectively understand the complex transmission process between variables, we further explored the relationship between external derivative variables and endogenous derivative variables for this complex behavior under the influencing factor network of NSSI.

### Factors selection method

Path analysis is an effective data analysis method that combines the advantages of human rational logic with the fast computing ability of the computer, in which the selection of variable factors is very important. Our research shows that individual factors such as weak physical condition, poor academic performance, tense peer relationships, and environmental factors such as embarrassing family conditions, low level of parental education, and poor family harmony may be potential risk factors for the participants to engage in NSSI behavior at the level of gender differences. Combined with the above single factor analysis and discussion results, we selected physical conditions (PC), academic performance (AP), peer relationship (PR), family conditions (FC), educational level of

**Table 4 Correlations among all of the main variables.**

|  | Average value | SD | RE | NCS | NSSI | PC | AP | PR | FC | ELP | FH |
|---|---|---|---|---|---|---|---|---|---|---|---|
| **Female** | | | | | | | | | | | |
| RE | 46.780 | 49.440 | 1 | | | | | | | | |
| NCS | −14.682 | 14.939 | 0.534** | 1 | | | | | | | |
| NSSI | 0.233 | 0.423 | 0.452** | 0.588** | 1 | | | | | | |
| PC | 2.713 | 0.518 | −0.114** | −0.138** | −0.169** | 1 | | | | | |
| AP | 2.495 | 0.836 | −0.047 | −0.093** | −0.086** | −0.018 | 1 | | | | |
| PR | 2.398 | 0.556 | −0.123** | −0.225** | −0.195** | 0.183** | −0.146** | 1 | | | |
| PC | 2.924 | 0.864 | 0.129** | 0.150** | 0.183** | −0.012 | −0.107** | 0.050 | 1 | | |
| ELP | 4.059 | 1.739 | −0.171** | −0.295** | −0.413** | 0.166** | −0.082** | 0.236** | 0.148** | 1 | |
| FH | 3.535 | 0.903 | −0.165** | −0.250** | −0.284** | 0.281** | −0.010 | 0.213** | 0.056* | 0.254** | 1 |
| **Male** | | | | | | | | | | | |
| RE | 51.492 | 45.387 | 1 | | | | | | | | |
| NCS | −18.213 | 12.617 | 0.477** | 1 | | | | | | | |
| NSSI | 0.067 | 0.251 | 0.212** | 0.276** | 1 | | | | | | |
| PC | 2.793 | 0.434 | −0.049 | −0.067* | −0.054 | 1 | | | | | |
| AP | 2.532 | 0.767 | −0.053 | −0.070* | −0.065* | −0.095** | 1 | | | | |
| PR | 2.462 | 0.514 | −0.028 | −0.017 | −0.047 | 0.116** | −0.178** | 1 | | | |
| PC | 2.869 | 0.858 | −0.020 | −0.009 | 0.053 | 0.078** | −0.120** | 0.114** | 1 | | |
| ELP | 4.525 | 1.624 | −0.045 | −0.062* | −0.117** | 0.079** | −0.175** | 0.123** | 0.289** | 1 | |
| FH | 3.687 | 0.765 | 0.026 | −0.059* | −0.075* | 0.109** | −0.072* | 0.076* | 0.122** | 0.114** | 1 |

**Note:**
* $p < 0.05$.
** $p < 0.01$.

parents (ELP), and family harmony (FH) as exogenous variables to consider the indirect impact on the NSSI behavior of the research object. As for endogenous variables, negative coping style (NCS) and radical education(RE) are proposed in this work and are used as endogenous derivative variables to uncover the relationship between external derivative variables and NSSI behavior. As shown in Table 4, the correlation analysis between all variables also supports the selection of model factors.

As shown in Table 4, From the female part, there is a significant correlation between the internal variables (EE and NCS) and the external variables (PC, AP, PR, FC, ELP, and FH). For males, the relationship between The significance is indeed reduced. In addition, both endogenous factors (extreme education, negative coping style) and NSSI of boys and girls have significant correlations, which suggests that endogenous factors have a significant impact on NSSI. There may be a mediating effect on the role, which provides some guidance for model construction.

### Construction of path model for NSSI behavior

Although the direct path between exogenous variables and NSSI is significant, there is also evidence that exogenous variables affect the occurrence of NSSI through endogenous variables, such as stressful life events, negative self-evaluation, drug use, and suicide
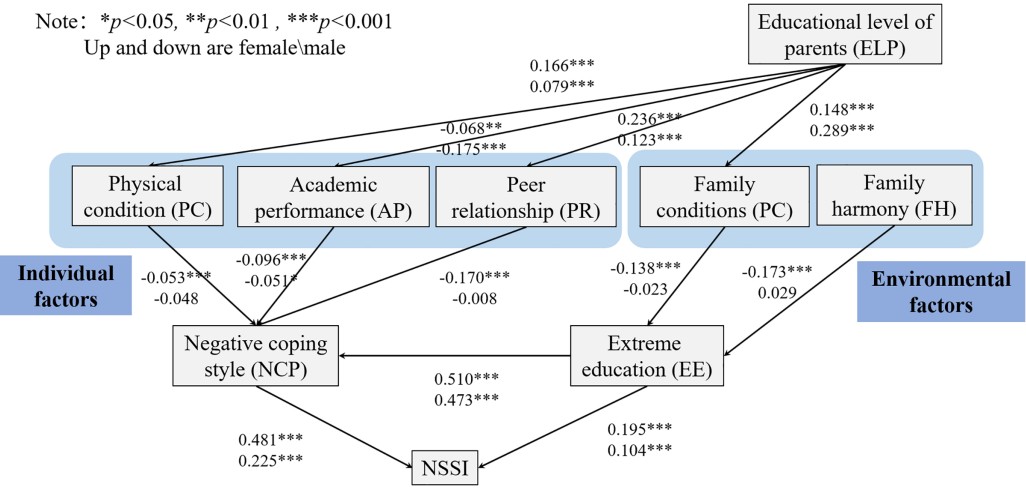

**Figure 1 Path diagram for the path analysis modeling of NSSI in gender difference.**

attempts. In the previous section, we completed a series of screening on the influencing factors with significant correlation and obtained the hint of a mediating effect between endogenous derivative variables. The above results were then incorporated into the subsequent path model construction to further explore the complex network of the intersection of external derivative variables and endogenous derivative variables of NSSI behavior. The construction results of the path network model are shown in Fig. 1. According to *Hu & Bentler (1999)*, several fitting indexes were used to evaluate the fitting effect of the model, including the comparative fitting index (CFI), the goodness of fit index (GFI), approximate root mean square error (SRMR) and weighted root mean square residual (RMSEA). Good model fitting is represented by the CFI value greater than 0.9, GFI value greater than 0.9, SRMR value less than 0.1, and RMSEA value less than 0.1. The model data fit well; the indicators of female and male part are: CFI = 0.913/0.923, GFI = 0.964/0.977, SRMR = 0.055/0.047, RMSEA = 0.097/0.069 with 90% confidence interval (CI) [0.084–0.111)/[0.054–0.084].

As shown in Fig. 1, the path relationship of each factor in the model is drawn accordingly. For females, (1) when radical education, peer relationship, academic achievement and physical condition affect negative coping style respectively, the standardized path coefficient values are 0.510 ($z = 17.454$, $P = 0.000 < 0.001$), −0.170 ($z = −7.814$, $P < 0.001$), −0.096 ($z = −4.425$, $P < 0.001$), −0.053 ($z = −2.436$, $P < 0.01$), which shows that radical education has a significant positive impact on negative coping style. Additionally, peer relationship, academic achievement, and physical condition have a significant negative impact on negative coping style. (2) When family harmony and family conditions have an impact on radical education, the standardized path coefficient values are −0.173 ($z = −9.542$, $P < 0.001$) and −0.138 ($z = 7.554$, $P < 0.001$), respectively. Therefore, it shows that family harmony will have a significant negative impact on radical education, and family conditions will have a significant positive impact on radical education. (3) When parents' education level affects family conditions, peer relationship,

academic performance, and physical condition, the standardized path coefficient values are 0.148 ($z = 5.862$, $P < 0.001$), 0.236 ($z = 9.518$, $P < 0.001$), −0.068 ($z = -3.227$, $P < 0.001$) and 0.166 ($z = 6.588$, $P < 0.001$) respectively, indicating that parents' education level affects family conditions, peer relationship, and physical conditions significantly. Additionally, parents' education level will have a significant negative impact on learning. (4) When negative coping style and radical education affect NSSI respectively, the standardized path coefficient values are 0.481 ($z = 20.636$, $P < 0.001$) and 0.195 ($z = 6.524$, $P < 0.01$) respectively, which indicates that negative coping style and radical education have a significant positive impact on NSSI.

For males, (1) when radical education, peer relationship, academic achievement, and physical condition affect negative coping style, the standardized path coefficient values are 0.473 ($z = 18.326$, $P < 0.001$), −0.008 ($z = -0.291$, $P = 0.771 > 0.05$), −0.051 ($z = -1.965$, $P = 0.049 < 0.05$), and −0.048 ($z = -1.847$, $P = 0.065 > 0.05$) respectively, which shows that radical education will have a significant positive impact on negative coping style, learning will have a significant negative impact on negative coping style, while peer relationship and physical condition will not have an impact on negative copying style. (2) When family harmony and family conditions affect radical education respectively, the paths are not significant ($z = 0.979$, $P = 0.328 > 0.05$), ($z = -0.798$, $P = 0.425 > 0.05$). Therefore, it shows that family harmony and family conditions do not affect radical education. (3) When parents' education level affects family conditions, peer relationship, academic performance and physical condition, the standardized path coefficient values are 0.289 ($z = 10.280$, $P < 0.001$), 0.123 ($z = 4.219$, $P < 0.001$), −0.175 ($z = -6.048$, $P = 0.000 < 0.01$), and 0.079 ($z = 2.679$, $P = 0.007 < 0.01$) respectively, indicating that parents' education level will affect family conditions respectively, peer relationship and physical condition have a significant positive impact, and parents' education level will have a significant negative impact on learning. (4) When negative coping style and radical education affect NSSI respectively, the standardized path coefficient values are 0.225 ($z = 7.057$, $P < 0.001$) and 0.104 ($z = 3.262$, $P < 0.001$), indicating that both negative coping style and radical education will have a significant positive impact on NSSI.

### Mediating effect of coping style in parental rearing style and NSSI

In the relationship between the independent variable and dependent variable, if the independent variable indirectly affects the dependent variable through a third variable, this behavior becomes an intermediary effect behavior. *Baron & Kenny (1986)* put forward the complete conceptual system and test procedure of an intermediary effect from the perspective of statistical methodology, which is generally accepted by researchers in the field of social sciences.

Based on the path model constructed above, this work will further discuss the mediating effects of endogenous variables such as parental radical education and negative coping styles, on NSSI behavior. Figure 2 shows a schematic diagram of the mediation model of NCS in the relationship between RE and NSSI. As shown in Fig. 2, the mediation model with negative coping methods as the intermediary variable, parental radical education as the independent variable, and NSSI as the dependent variable component are significant

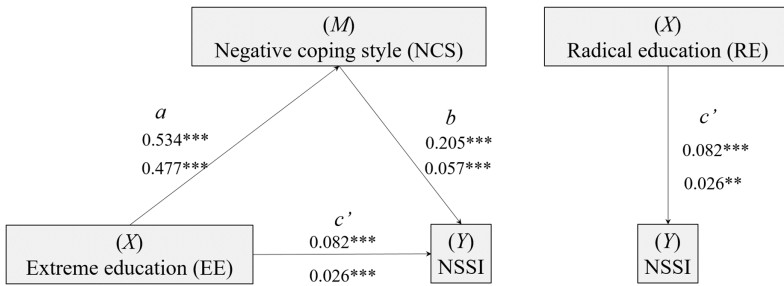

Note:**P<0.01 , ***P<0.001
Up and down are female\male

**Figure 2 The mediating model of negative coping style (NCS) in the relationship between RE and NSSI.**

**Table 5 Summary of mediation test results.**

| | Test conclusion | c | a | b | a*b | c′ | a*b (95% BootCI) | Effect proportion (a*b/c) |
|---|---|---|---|---|---|---|---|---|
| Female | | | | | | | | |
| RE→ NCS→NSSI | Partial intermediary | 0.191 *** | 0.534 *** | 0.205 *** | 0.109 | 0.082 *** | 0.229–0.290 | 57.191% |
| Male | | | | | | | | |
| RE→ NCS→NSSI | Partial intermediary | 0.053 *** | 0.477 *** | 0.057 *** | 0.027 | 0.026 ** | 0.076–0.140 | 50.879% |

Note:
** $p < 0.01$.
*** $p < 0.001$.

for the girls in the study. The indirect impact of parental radical education is 0.109 ($SE = 0.016$, 95% CI [0.172–0.210]). In addition, before using negative coping methods as media, the regression coefficient of the direct relationship between parental radical education and NSSI was significant ($B = 0.191$, 95% CI [0.172–0.210]). When negative coping methods were added to the analysis, the correlation between parental radical education and NSSI was still significant ($B = 0.082$, 95% CI [0.062–0.102]). This shows that negative coping methods play a partial mediating role in the relationship between parental radical education and NSSI. Negative coping methods accounted for 57.2% of the total impact of radical education on NSSI. See Table 2 for specific parameters. The intermediary model is also significant for the boys in the study. The indirect impact of parental radical education is 0.027 ($SE = 0.016$, 95% CI [0.076–0.140]). In addition, before using negative coping methods as media, the regression coefficient of the direct relationship between parental radical education and NSSI was significant ($B = 0.053$, 95% CI [0.039–0.067]). When negative coping methods were added to the analysis, the correlation between parental radical education and NSSI was still significant ($B = 0.026$, 95% CI [0.010–0.042]), indicating that negative coping methods played a partial intermediary role in the relationship between radical education and NSSI. Negative coping methods account for 50.872% of the total impact of parental radical education on NSSI. See Table 5 for specific parameters.

In Table 5, *c* represents the regression coefficient of independent variable *x* to dependent variable *y* (when there is no intermediary variable *m* in the model), that is, the total effect. *a* represents the regression coefficient of *x* to *m*, *b* represents the regression coefficient of *m* to *y*, and *a* * *b* is the product of *a* and *b*, that is, the intermediary effect. *c′* represents the regression coefficient of *x* to *y* (when there is an intermediary variable *m* in the model), that is, the direct effect. 95% BootCI refers to the 95% confidence interval calculated by bootstrap sampling. If the interval does not include 0, it indicates that the intermediate effect is significant.

As shown in Table 5, regulated by gender differences, after introducing the intermediary variable *m* (*i.e.*, negative coping style), the direct effect (*i.e.*, regression coefficient) of radical education on NSSI shows a downward trend (female: 0.191, 0.082; male: 0.053, 0.026), which shows that the relationship between RE and NSSI is further constructed through the intermediary variable negative coping style. The effects of women and men accounted for 57.191% and 50.879%, both of which were partial mediating effects. At the same time, the 95% confidence intervals calculated by bootstrap sampling do not cover 0, which verifies that our model has a significant mediating effect.

## DISCUSSION

The high prevalence and harmfulness of NSSI have made this behavior a serious public health concern worldwide. In this study, a total of 2,689 valid responses (43.03% and 56.97% for males and females, respectively) were collected, with a positive rate of 15.16% for NSSI behavior. The detection rate of NSSI among adolescents in China ranged from 5.44% to 23.2%, as reported in a previous study (*Xu et al., 2020*), which is consistent with our data. At the same time, we found that NSSI behavior showed gender differences: 357 females had NSSI behaviors compared with 78 males, and the percentage of female students with NSSI behaviors (23.30%) was significantly higher than that of male students (6.74%). While some studies have reported no difference in the prevalence of NSSI between men and women (*Swannell et al., 2014*; *Klonsky, Oltmanns & Turkheimer, 2003*; *Nock et al., 2006*), others have shown that the prevalence of NSSI is significantly higher in women due to multiple physiological, psychological, and social factors (*Barrocas et al., 2012*; *Sornberger et al., 2012*). Studies by *Wilkinson et al. (2022)* (*Plener et al., 2015*) have shown that the effect of gender on NSSI is age-related and more significant in adolescence. Our investigation into an adolescent population showed that NSSI gender difference results were consistent with this feature. Studies have shown that those with NSSI behaviors rarely adopt positive coping styles such as problem-solving, regulating negative emotions, or seeking help, while most have negative coping styles such as substance abuse, avoidance, and self-blaming denial (*Williams & Hasking, 2010*; *Cawood & Huprich, 2011*; *Kiekens et al., 2015*; *Castro & Kirchner, 2018*). The family parenting style is also a major factor influencing adolescents' emotion management and coping styles and even NSSI behaviors (*Wei et al., 2022*). *Ying et al. (2021)* found that parents' use of harsh punishment, excessive control, and denial of negative parenting were positively associated with adolescent NSSI, while a review by *Fong et al. (2022)* found that low parental psychological support, high psychological control, and reactive control were highly correlated with NSSI.

Thus, in examining NSSI, the relationship between parenting styles, secondary school students' coping styles, and emotion management may contribute to NSSI through multiple pathways. To this end, this work examines the interaction between the parenting styles boys and girls receive, the ways they cope with problems, and the behaviors that produce NSSI.

From our work, we found that the current risk factors for adolescent NSSI are complex and include a variety of factors including the individual, the school, and the family. Therefore, a multifaceted study of the basic characteristics of the subjects was carried out in the initial phase of this work. Two factors were screened out as they were not statistically significant: parental expectations and parental commitment to the child. At the same time, we found that frailty was a risk factor for NSSI behavior in secondary school students in terms of physical condition. This might be because frail secondary school students are more likely to adopt emotionally directed coping styles such as patience, avoidance, and fantasy denial. The higher the *per capita* monthly household income, the higher the prevalence of NSSI among secondary school students, perhaps because in affluent family environments, parents show significantly high demands and strong interference with their children, which to some extent can induce NSSI behaviors in children. In terms of parenting style, upbringing is one of the reference standards for their children's education, and thus the reference standards of more educated parents tend to be more rational and reasonable. As shown in Table 1, the prevalence of NSSI among secondary school students tended to decrease gradually as parents' education level increased, but in particular, we found that the prevalence of NSSI among secondary school students increased slightly when parents' education level reached university and above. Compared with parents with higher education levels, parents with lower education levels (primary school) tend to adopt negative parenting styles such as harsh punishment and denial, which make it difficult to provide rational and effective guidance for children's confusion and problems during adolescence. In contrast, when parents reach a certain level of education (university and above), this positive effect changes to a negative one, probably because the reference standard is too high, putting more stringent demands on the child in life and learning, and adopting more negative parenting styles such as over-interference and control (*Xiang et al., 2020*), which are not conducive to the development of a mature and positive coping style. Negative parenting styles tend to impair socioemotional development and emotional management skills, thus may lead to adolescents engaging in psychological and behavioral avoidance through NSSI behaviors. The path model constructed in this work verifies these inferences to some extent.

**Parenting style and NSSI**

Significance test analysis was conducted using an independent sample t-test on the parenting style of the participants on NSSI behavior and gender respectively (Table 2). It was found that the parenting styles of the adolescents were extreme and contradictory: parents treat their children with excessive severity on the one hand and excessive care on the other, which is consistent with the study by *Zuo, Xi & Sang (2004)*. Therefore, this type of parenting style is classified as "radical education" in this study. The boundaries of

"radical education" are unclear, extreme, and full of contradictions, which on the one hand directly increases children's stress and makes them more likely to develop tension and on the other hand, will impair children's emotional control, which will make children have a negative tendency to cope with stress. Together, these two pathways put children at increased risk of developing NSSI. In contrast, emotional warmth and understanding shown by parents can increase their children's psychological and emotional resilience (*Guerreiro et al., 2013*), thus reducing the risk of their children developing NSSI. At the same time, studies combining family environment factors (see Table 1) observed that families with NSSI also generally have higher levels of conflict and lower levels of relaxed recreation (*Domínguez-Baleón et al., 2018*; *Liu et al., 2020*). In the traditional Chinese parenting concept, parents have a relatively absolute say in the family, and their parenting style has an inevitable and profound impact on the physical and mental health of their children (*Gu, Ma & Xia, 2020*; *Serafini et al., 2017*). When a child's behavior or emotions are problematic, the response of the parents often determines the outcome. If parents adopt a punitive parenting style and emotionally deny and reject, disruptions in the child's cognitive ability may happen (*Masud et al., 2019*). At the same time, from a physiological point of view: the disruption of the level of secretion of transmitters such as norepinephrine in the adolescent's body during the reception of negative information input makes it impossible to inhibit the response to the stimulus until it is adequately evaluated, *i.e.*, the rapid response impulse, which eventually leads to risky behavior (*Swann et al., 2013*; *Francis et al., 1999*) and even to the development of NSSI. In particular, we found an interesting phenomenon: gender differences present a clear difference in the significance level of the effects of parenting styles. Both positive parenting style factor scores and negative parenting behavior factor scores were greater for boys than for girls, implying that compared to girls, boys faced more positive parenting behaviors in family relationships while suffering more negative parenting styles, including punitive severity and denial. In China, there is still a certain degree of "son preference" in education, and boys are often expected to grow up with more responsibility, so they are more involved in family education. In the initial stage of family education, positive inductive education such as emotional warmth and understanding is often reflected (*Liu, Su & Yin, 2022*) but when there is a certain gap between expectations and reality or when the child shows resistant behavior, such caring education will gradually transform into radical education in the form of severe punishment, denial, and excessive interference, to further realize the parents' expectations. At the same time, in some Chinese families with strong traditional values, the parenting style of girls does not have this contradictory situation of "high expectations and high strictness" (*Hannum, Kong & Zhang, 2009*; *Ong et al., 2018*). In conclusion, the gender salience of parenting styles is closely related to the unique Chinese social context. An in-depth analysis of this phenomenon provides a theoretical basis for effectively avoiding gender-specific NSSI behaviors. In conclusion, the gender salience of parenting styles is closely related to the unique Chinese social context.

## Coping style and NSSI

As shown in Table 3, we found that positive problem-oriented coping styles (seeking social support, positive rationalization) were protective factors for NSSI while negative emotion-oriented coping styles (tolerance, avoidance, fantasy denial) were risk factors for NSSI. In a review, *Fu et al. (2018)* similarly found that people with NSSI behaviors were more inclined to fantasy avoidance, self-blame, venting, patience, substance abuse, and other negative coping styles directed toward emotions. In this study, NSSI behaviors were highly correlated with the coping styles of secondary school students, and the NSSI group tended to adopt patience, avoidance, and denial in coping with individual and external stressors, which tended to cause the accumulation of negative emotions within and ultimately lead to poor outcomes.

Interestingly, there are obvious differences in the handling of stress as well as negative emotions among different genders: boys are more inclined to adopt a positive confrontation and reasonable solution, while girls are more inclined to make coping behaviors that are more emotional such as avoidance, venting emotions, and fantasy denial. For a long time, with the development of social productivity and the emergence of male prominence in the production sector, men began to dominate, *i.e.*, patriarchal society gradually became the mainstream in the development of human history nowadays. The patriarchal culture of "male superiority and female inferiority" has had a profound impact on the difference in social status between men and women, resulting in men being given or expected to have personality qualities such as strength, independence, and extroversion, while women are usually given or expected to have personality qualities such as demureness and introversion (*Zosuls et al., 2011*). Established research on personality and gender roles has shown that masculinity scores have a strong positive correlation with the extraversion dimension, and femininity scores have a strong positive correlation with the interpersonal dimension, with cross-cultural consistency (*Schmitt et al., 2017*). Therefore, in terms of emotion regulation strategies, girls adopt emotion-focused strategies while boys tend to use cognitive reappraisal strategies. The difference in strategies may lead to less effective regulation in girls than in boys, which can easily trigger adverse emotions such as anxiety and depression and lead to more psychological problems and more self-injurious behaviors.

## Exogenous and endogenous derivative variables on NSSI

NSSI is in essence a maladaptive coping strategy that can be interpreted as a way of regulating tension. Studies have shown (*Xu et al., 2021*) that NSSI behaviors are not directly caused by a single factor, but are often influenced by the interaction of social and environmental factors, family factors, and personal factors in the process of their formation. In particular, there seems to be an interesting cross-influence phenomenon among the influencing factors, and this cross-influence relationship is accompanied by either indirect or direct extrapolation and endogeneity (*i.e.*, the role of exogenous and endogenous variables). Ecosystem theory suggests that personality traits, endogenous variables, and exogenous environmental variables jointly influence adolescents' regulation (*Panlilio, Jones Harden & Harring, 2018*). That is, emotion regulation is the result of the

interaction between the individual and the external environment. Parenting styles and adolescents' coping styles do not seem to be independent events: children with positive parenting styles receive more support and trust when facing problems, have more courage to face life challenges, apply resources positively, solve problems, and thus are better able to regulate stressful emotions under pressure. On the contrary, children with a negative parenting style often have a sense of mistrust, are not confident enough to face problems, and often exhibit negative coping behaviors such as procrastination and social detachment, which lead to psychological problems such as obvious NSSI symptoms.

To this end, this work constructs a pathway analysis model of multiple influences on NSSI behavior to effectively explore the extrapolative and endogenous variables among the influences on this complex behavior. In terms of the influence of endogenous variables on NSSI, we found that negative coping styles mediated the association between radical education and NSSI behaviors in the pathway model. Positive parenting with warm and understanding parents is a major contributor to the development of high self-esteem, effective emotion regulation management, and problem-oriented positive coping styles in adolescents, while negative and radical parenting styles such as harsh punishment, denial, and excessive control are responsible for the development of immature and negative coping styles in adolescents, thus making them more likely to trigger NSSI behaviors in the face of external stimuli (*Liu et al., 2021*; *Peng, Fu & Zhang, 2020*). On the other hand, adolescents who are exposed to long-term negative parenting styles often have difficulties in receiving effective communication and help from their parents, and tend to adopt more negative coping styles that gravitate towards emotions (patience, avoidance, venting emotions, fantasy/denial) when dealing with stress alone, which gradually evolve into some kind of self-harm phenomenon (*i.e.*, NSSI behavior) when these coping styles are not effective. Therefore, it is necessary to adopt positive (warmth, understanding) parenting styles and suppress negative (punishment, harshness, rejection, denial) parenting styles to reduce the immature coping styles of middle school students.

As shown in Fig. 2, the contribution factor of the mediating effect in gender differences showed a significant difference: in the face of radical family education, females were more likely to adopt a negative coping style for diversion (female effect coefficient: 0.534, $P < 0.05$; male effect coefficient: 0.477, $P < 0.05$); at the same time, negative coping style showed a more significant influence on females' NSSI behavior compared to that of males (female effect coefficient: 0.205, $P < 0.05$; male effect coefficient: 0.057, $P < 0.05$). The role of negative coping styles showed a more significant effect on women's NSSI behavior compared to men's mediation effect (female effect coefficient: 0.205, $P < 0.05$; male effect coefficient: 0.057, $P < 0.05$). Although the current findings advanced the understanding that the existence of gender differences in NSSI, the mechanism of gender differences' influence on NSSI behavior is currently less studied. It has been found that the gendered socialization of emotions, *e.g.*, shame and anger, may influence the types of emotions experienced by men and women, leading to women being more likely to engage in NSSI (*Domes et al., 2010*; *Buchanan et al., 2010*), and that this gendered socialization of emotions is often closely related due to the differential levels of perceptions of the status roles of men and women in society at large. At the same time, biological factors such as
hormonal differences between males and females (*e.g.*, testosterone and estradiol) also seem to be involved in gender-influenced relationships in NSSI.

Here, a noteworthy finding was that females used emotionally directed negative coping more frequently than boys in conflict resolution and scored higher on all behaviors of negative coping, while good peer relationships had a significant protective effect on girls. The effect of individual factors on boys' coping styles was not significant. In addition, girls' negative coping behaviors were more influenced by parenting styles than were boys'. These results suggest that girls may be more prone to mood swings in the face of stressful conflicts, leading them to consistently use more negative coping styles than boys to the point of developing more mental health problems, which may imply a greater risk of developing NSSI. Another noteworthy finding is that the way girls are educated is significantly influenced by their family environment; a harmonious family atmosphere has a significant negative effect on aggressive parenting styles, and good family conditions have a positive effect, presumably related to a higher *per capita* monthly household income and more intrusive parental control over children.

From the perspective of extrapolation quantity, the parental education level influences the NSSI phenomenon in various ways, and it affects both individual and environmental factors of the child, and ultimately affects the generation of NSSI behavior from different paths, respectively. Therefore, in general, only secondary school students whose parents' education level is elementary school have the highest incidence of NSSI, and as the parental education level increases, the incidence of NSSI decreases. However, the incidence of NSSI increases for parents with university education level and above; parents with low education level (elementary school) are likely to use more negative parenting styles such as harsh punishment and denial, while parents with higher education put high demands on their children and use more negative parenting styles such as excessive interference and control (*Zhang, Cheng & Xiao, 2014*), both of which can impair children's development of mature, positive coping styles; socioemotional development, and emotional management skills. In addition, there is a positive relationship between parental education and family income, and family socioeconomic status is a reflection of the level of resources available to the individual and the family. Individuals with low family socioeconomic status not only have higher levels of economic stress but also face more uncontrollable stress. They also face more uncontrollable negative life events and persistent stressful experiences. Society also has negative perceptions and judgments of individuals with low family socioeconomic status or low income (*e.g.*, they are generally perceived as lazy and unmotivated), and these negative stereotypes or prejudices are risk factors for individuals' self-esteem levels. At the same time, these stereotypes or prejudices can also cause individuals of low family socioeconomic status to have more experiences of social exclusion (*i.e.*, being ignored or rejected by others), which is not conducive to children forming good peer relationships. However, parental education showed an inverse relationship with academic achievement, which is different from the conventional perception that highly qualified parents place more importance on their children's education but is consistent with the phenomenon of "poor students" in China. Students whose families have a weaker financial base are likely to make more efforts to change their *status quo*, and studying is the best way for them to do

so. However, in this state, they are also prone to develop a focused and withdrawn personality, which is not conducive to emotional regulation and stress release, thus increasing the risk of NSSI.

## CONCLUSION

In summary, multiple factors influenced the occurrence of NSSI. A negative coping style directed toward emotions of tolerance, avoidance, and fantasy denial increased the occurrence of NSSI in secondary school students, while a positive coping style directed toward solving problems, seeking social support and positive rationalizing explanation reduced the occurrence of NSSI, and the parenting style could indirectly influence the occurrence of NSSI by affecting the coping style of secondary school students. Family parenting style and adolescent coping style also showed significant gender differences in the occurrence of NSSI. Males were the target of both positive and negative parenting styles, while females, who were more likely to choose emotionally directed negative coping styles in response to external stimuli, showed a more significant predisposition for NSSI behavior. This phenomenon seems to be influenced by multi-level factors such as sociocultural, individual value identity, and physiological structure differences. In the path analysis model with the introduction of mediating effects, the influence of gender differences on NSSI behavior becomes more pronounced under the interaction of multiple factors: women seem to be more significantly influenced by external variables than did male subjects, and are more likely to trigger NSSI behavior under the interaction of multiple factors.

Therefore, an important entry point for future NSSI interventions for secondary school students can start from the following levels: (1) Parents of secondary school students should focus on communication and understanding with their children *e.g.*, understanding, respecting, and affirming their children's thoughts and behaviors, and building solutions together for the difficulties they encounter in their academic life. Parents ought to create a harmonious family atmosphere, pay attention to the children's psychological state, and guide them to form good psychological characteristics.
For children with NSSI tendencies, dangerous objects such as medicines, knives, and glass should be strictly managed. If NSSI occurs frequently, the child should be sent to a specialist hospital in time. At the community, school, and government levels: adolescent mental health should be given attention, and supervision of undesirable factors that may lead to NSSI-related behaviors should be strengthened. Cracking down on social phenomena that may lead to NSSI-related behaviors, as well as strengthening students' physical exercise and guiding them to form a good social and learning atmosphere will also reduce the occurrence of NSSI to a certain extent. (2) At the level of gender differences: pay attention to the influence of family education and coping styles on different gender targets. Compared with males, female groups are more susceptible to the influence of externally derived factors, and it is more important to adopt an encouragement-oriented positive family education style and behavior-oriented coping style to avoid such negative, denial-avoidance, and other undesirable emotions.

## STRENGTHS AND LIMITATIONS

In terms of strengths, the results of the present study are based on a relatively large sample drawn from a randomized whole group and can be representative of the NSSI characteristics of secondary school students in Zhejiang Province, China, and to some extent other regions and cultures with potential public health problems with NSSI. The present results may provide a reference for some countries with similar cultural backgrounds.

In terms of limitations, first of all, NSSI is a heterogeneous condition that encompasses a wide range of behaviors with different clinical correlates. Furthermore, although the most commonly used measure in most studies explicitly instructs participants to "answer yes to the question only if you intentionally or deliberately hurt yourself," other measures are less clear about the behavior for which the question is intended to cause harm. Thus, these scales may not capture some atypical behaviors. Second, we did not examine gender differences in the severity and duration of the NSSI. This study is a cross-sectional survey with limitations for longitudinal judgments of NSSI development; so, longitudinal prospective observations may be considered in the future. Therefore, whether men or women have a more severe or chronic NSSI remains undetermined.

We hope that this study will inspire other researchers to develop new ideas in an attempt to understand the diversity of sex-differentiated studies in NSSI. Identifying these (multiple dimensions of gender variability) will help advance new theories and perhaps eventually inform gender-specific prevention and intervention efforts aimed at reducing NSSI.

### Funding

This work was supported by the Ningbo Health Branding Subject Fund (PPXK2018-08), the Ningbo Public Welfare Technology Project (20211JCGY020281), the Zhejiang Medical and Health Science and Technology (2021KY1063, 2021KY330), and the Ningbo Natural Science Foundation (202003N4262). The funders had no role in study design, data collection and analysis, decision to publish, or preparation of the manuscript.

### Grant Disclosures

The following grant information was disclosed by the authors:
Ningbo Health Branding Subject Fund: PPXK2018-08.
Ningbo Public Welfare Technology Project: 20211JCGY020281.
Zhejiang Medical and Health Science and Technology: 2021KY1063, 2021KY330.
Ningbo Natural Science Foundation: 202003N4262.

### Competing Interests

The authors declare that they have no competing interests.

## Author Contributions

- Fang Cheng analyzed the data, prepared figures and/or tables, authored or reviewed drafts of the article, and approved the final draft.
- Changzhou Hu analyzed the data, authored or reviewed drafts of the article, and approved the final draft.
- Wenwu Zhang conceived and designed the experiments, prepared figures and/or tables, and approved the final draft.
- Huabing Xie conceived and designed the experiments, prepared figures and/or tables, and approved the final draft.
- Liangliang Shen analyzed the data, authored or reviewed drafts of the article, and approved the final draft.
- Beini Wang conceived and designed the experiments, authored or reviewed drafts of the article, and approved the final draft.
- Zhenyu Hu performed the experiments, authored or reviewed drafts of the article, and approved the final draft.
- Yucheng Wang performed the experiments, analyzed the data, authored or reviewed drafts of the article, and approved the final draft.
- Haihang Yu conceived and designed the experiments, performed the experiments, prepared figures and/or tables, and approved the final draft.

## Human Ethics

The following information was supplied relating to ethical approvals (*i.e.*, approving body and any reference numbers):

The study was approved by the ethics committee of Ningbo Corning hospital (approval No.: nbknyy-2020-lc-52), and all research investigation methods carried out in this work were conducted in accordance with the relevant guidelines and regulations in the declaration of Helsinki.

## Data Availability

The raw data is available in the Supplemental File.

## Supplemental Information

Supplemental information for this article can be found online at http://dx.doi.org/10.7717/peerj.14507#supplemental-information.

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
