# Peer review of "The influence of parenting style and coping behavior on nonsuicidal self-injury behavior in different genders based on path analysis"

_PeerJ, doi:10.7717/peerj.14507_

## Round 0.1 · original submission · Minor Revisions

I have now received the reviewers' comments on your manuscript. They have suggested some minor revisions to your manuscript. Therefore, I invite you to respond to the reviewers' comments and revise your manuscript.

Reviewer 1 ·

Basic reporting

The researcher writes the results of his research report in a complete and structured manner. Writing in English which is easy to understand. The research background is in accordance with the objectives to be achieved. The display of research data is presented in an attractive manner and according to the rules of correct placement. The conclusion shows that this research is new and very good to spark the next research.

Experimental design

The research objectives are conveyed properly and correctly. The research question has attempted to address discrepancies from other studies. The method is carried out carefully, in detail and directed, but has not yet presented a review of research ethics.

Validity of the findings

This research can be said to be quite interesting and renewable because it shows an increasing phenomenon that occurs in the age group that is currently prone to self-harm behavior. Research results are reported in detail and show a strong relationship with what is contained in the research question. The conclusions presented are very appropriate in answering the issues that develop in the research background.

Additional comments

A complete explanation of the passage of the research ethics review should be explained in this research paper.

Reviewer 2 ·

Basic reporting

Thanks to the respected authors, a very valuable and important research has been done in this field and has been written with precision and detail. There are some subtle points that please correct;
Keywords are not very suitable except for the first keyword.
The abstract is not structured. It is better to rewrite it in the form of background, materials and methods, results and conclusions.
The type of study, sampling method, formula for calculating the sample size should be mentioned.
The section related to the results of statistical analysis should be separated under the title "Results".

Experimental design

no comment

Validity of the findings

no comment

·

Basic reporting

English language is acceptable.
literature review is sufficient.
Please, clearly state aim and objectives, in sub-section as Current Study or Purpose of the current study. Current Study, It should be explicitly stated for the reader how the present study makes a unique contribution to the literature.

Experimental design

Add the information related validity or reliability of the measure in your study.
Please report effect size.

Validity of the findings

Conclusions are well stated, linked to original research question & limited to supporting results.

Additional comments

there is well-written document.
the figure must be re-draw.
DOI should be added for references.

---

## Round 0.2 · Minor Revisions

Many thanks for submitting the revised manuscript in PeerJ journal. Based on one reviewer's opinion, this version of article needs some minor revisions. I invite you responding carefully to the reviewer's comments.

Reviewer 1 ·

Basic reporting

The researcher submits the report writing in a complete and systematic manner. The use of English can be said to be quite good and easy to understand. The background of the research is in accordance with what the researcher will do. Supporting data has been presented in full with the use of open tables. The hypothesis can produce a new concept and model related to the phenomenon raised in this study.

Experimental design

The aims and objectives of the study are an explanation in responding to the phenomenon of self-harm which has recently increased in incidence among adolescents, especially in urban areas. Research questions are clearly stated and the researcher tries to reveal any knowledge gaps based on other research that has been done.

Validity of the findings

This study is sufficient to explain the statistical relationship of the research findings. The conclusion of the study is in accordance with what is contained in the research question. The researcher has also conveyed the limitations of the study as an interesting discussion.

Additional comments

Research can be said to be very worthy to be published because it tries to answer the psychosocial aspects of the latest phenomenon that is developing in the adolescent population. My suggestion is that the researcher should correct the written statements in lines starting from 271 to 277. In addition, also on lines 278 to 282. because I think there are repetitions of words that need to be corrected again.

·

Basic reporting

The revision manuscript can be published in PEER J.

Experimental design

No Cm.

Validity of the findings

No Cm

Additional comments

No Cm

---

## Round 0.3 · accepted · Accept

Many thanks for addressing all the issues.

Reviewer 1 ·

Basic reporting

The researcher has presented the introduction and sequence of literature review in a structured and systematic manner according to the suggested suggestions for improvement.

Experimental design

Researchers have improved the revision of the paper well so that readers can easily understand the research topic raised

Validity of the findings

This research is interesting by displaying appropriate previous research and a complete literature review according to the research topic

Additional comments

This research is good and interesting to be published